# Towards Understanding Fast Adversarial Training

## Abstract

Current neural-network-based classifiers are susceptible to adversarial examples. The most empirically successful approach to defending against such adversarial examples is adversarial training, which incorporates a strong self-attack during training to enhance its robustness. This approach, however, is computationally expensive and hence is hard to scale up. A recent work, called fast adversarial training, has shown that it is possible to markedly reduce computation time without sacrificing significant performance. This approach incorporates simple self-attacks, yet it can only run for a limited number of training epochs, resulting in sub-optimal performance. In this paper, we conduct experiments to understand the behavior of fast adversarial training and show the key to its success is the ability to recover from overfitting to weak attacks. We then extend our findings to improve fast adversarial training, demonstrating superior robust accuracy to strong adversarial training, with much-reduced training time.

## 1 Introduction

Adversarial examples are carefully crafted versions of the original data that successfully mislead a classifier (Szegedy et al., 2013), while realizing minimal change in appearance when viewed by most humans. Although deep neural networks have achieved impressive success on a variety of challenging machine learning tasks, the existence of such adversarial examples has hindered the application of deep neural networks and drawn great attention in the deep-learning community.

Empirically, the most successful defense thus far is based on Projected Gradient Descent (PGD) adversarial training (Goodfellow et al., 2014; Madry et al., 2017), augmenting the data of interest with strong adversarial examples, to help improve model robustness. Although effective, this approach is not efficient and may take multiple days to train a moderately large model. On the other hand, one of the early versions of adversarial training, based on a weaker Fast Gradient Signed Method (FGSM) attack, is much more efficient but suffers from "catastrophic overfitting," a phenomenon where the robust accuracy with respect to strong attacks suddenly drops to almost zero during training (Tramèr et al., 2017; Wong et al., 2019), and fails to provide robustness against strong attacks.

Fast adversarial training (Wong et al., 2019) is a simple modification to FGSM, that mitigates this issue. By initializing FGSM attacks with large randomized perturbations, it can efficiently obtain robust models against strong attacks. Although the modification is simple, the underlying reason for its success remains unclear. Moreover, fast adversarial training is only compatible with a cyclic learning rate schedule (Smith & Topin, 2019), with a limited number of training epochs, resulting in sub-optimal robust accuracy compared to PGD adversarial training (Rice et al., 2020). When fast adversarial training runs for a large number of epochs, it still suffers from catastrophic overfitting, similar to vanilla FGSM adversarial training. Therefore, it remains an unfinished task to obtain the effectiveness of PGD adversarial training and the efficiency of FGSM adversarial training simultaneously.

In this paper, we conduct experiments to show that the key to the success of fast adversarial training is not avoiding catastrophic overfitting, but being able to retain the robustness of the model when catastrophic overfitting occurs. We then utilize this understanding to propose a simple fix to fast adversarial training, making possible the training of it for a large number of epochs, without sacrificing efficiency. We demonstrate that, as a result, we yield improved performance.

We also revisit a previously developed technique, FGSM adversarial training as a warmup (Wang et al., 2019), and combine it with our training strategy to further improve performance with small additional computational overhead. The resulting method outperforms the state-of-the-art approach, PGD adversarial training (Rice et al., 2020), while consuming much less training time.

Our contributions are summarized as follows:

- We conduct experiments to explain both the success and the failure of fast adversarial training for various cases.
- We propose an alternative training strategy as a fix to fast adversarial training, which is equivalently efficient but allows training for a large number of epochs, and hence achieves better performance.
- We propose to utilize the improved fast adversarial training as a warmup for PGD adversarial training, to outperform the state-of-the-art adversarial robustness, with reduced computation.

## 2 BACKGROUND AND RELATED WORK

The existence of adversarial examples in deep learning was initially reported in (Szegedy et al., 2013). Since then, many approaches have been proposed to mitigate this issue and improve the adversarial robustness of models. A straightforward method is data augmentation, where adversarial examples are generated before the back-propagation at each iteration and used for model updates. This approach is referred to as adversarial training. It was first used with a gradient-based single-step adversarial attack, also known as the Fast Gradient Sign Method (FGSM) (Goodfellow et al., 2014). Later, (Kurakin et al., 2016) found that models trained with FGSM tend to overfit and remain vulnerable to stronger attacks. They proposed a multi-step version of FGSM, namely the Basic Iterative Method (BIM), seeking to address its weaknesses. Randomized initialization for FGSM then was introduced in (Tramèr et al., 2017), leading to R+FGSM to increase the diversity of attacks and mitigate the overfitting issue. Finally, (Madry et al., 2017) combined randomized initialization with multi-step attacks to propose projected gradient descent (PGD) attacks, and showed its corresponding adversarial training is able to provide strong adversarial robustness (Athalye et al., 2018). As PGD adversarial training is effective, many works have tried to improve upon it (Zhang et al., 2019b; Xie et al., 2019). However, a recent study (Rice et al., 2020) conducted extensive experiments on adversarially trained models and demonstrated that the performance gain from almost all recently proposed algorithmic modifications to PGD adversarial training is no better than a simple piecewise learning rate schedule and early stopping to prevent overfitting.

In addition to adversarial training, a great number of adversarial defenses have been proposed, yet most remain vulnerable to stronger attacks (Goodfellow et al., 2014; Moosavi-Dezfooli et al., 2016; Papernot et al., 2016; Kurakin et al., 2016; Carlini & Wagner, 2017; Brendel et al., 2017; Athalye et al., 2018). A major drawback of many defensive models is that they are heuristic and vulnerable to adaptive attacks that are specifically designed for breaking them (Carlini et al., 2019; Tramer et al., 2020). To address this concern, many works have focused on providing provable/certified robustness of deep neural networks (Hein & Andriushchenko, 2017; Raghunathan et al., 2018; Kolter & Wong, 2017; Weng et al., 2018; Zhang et al., 2018; Dvijotham et al., 2018; Wong et al., 2018; Wang et al., 2018; Lecuyer et al., 2018; Li et al., 2019; Cohen et al., 2019), yet their certifiable robustness cannot match the empirical robustness obtained by adversarial training.

Among all adversarial defenses that claim empirical adversarial robustness, PGD adversarial training has stood the test of time. The only major caveat to PGD adversarial training is its computational cost, due to the iterative attacks at each training step. Many recent works try to reduce the computational overhead of PGD adversarial training. (Shafahi et al., 2019) proposes to update adversarial perturbations and model parameters simultaneously. By performing multiple updates on the same batch, it is possible to imitate PGD adversarial training with accelerated training speed. Redundant calculations are removed in (Zhang et al., 2019a) during back-propagation for constructing adversarial examples, to reduce computational overhead. Recently, (Wong et al., 2019) shows surprising results that FGSM adversarial training can obtain strongly robust models if a large randomized initialization is used for FGSM attacks. However, they are forced to use a cyclic learning rate schedule (Micikevicius et al., 2017) and a small number of epochs for the training. This issue limits its performance, especially when compared to state-of-the-art PGD adversarial training with early stopping (Rice et al., 2020).

## 3 FAST ADVERSARIAL TRAINING

### 3.1 PRELIMINARIES

We consider the task of classification over samples $(x, y) \in (\mathcal{X}, \mathcal{Y})$. Consider a classifier $f_\theta : \mathcal{X} \to \mathcal{Y}$ parameterized by $\theta$, and a loss function $\mathcal{L}$. For a natural example $x \in \mathcal{X}$, an adversarial example $x'$ satisfies $\mathcal{D}(x, x') < \epsilon$ for a small $\epsilon > 0$, and $f_\theta(x) \neq f_\theta(x')$, where $\mathcal{D}(\cdot, \cdot)$ is some distance metric, *i.e.*, $x'$ is close to $x$ but yields a different classification result. The distance is often described in terms of an $\ell_p$ metric, and we focus on the $\ell_\infty$ metric in this paper.

Adversarial training is an approach for training a robust model against adversarial attacks. It represents the objective of obtaining adversarial robustness in terms of a robust optimization problem, defined as

$$\min_\theta E_{(x,y) \sim \mathcal{X}} \max_{\|x'-x\|_\infty < \epsilon} (\mathcal{L}(f_\theta(x'), y)) \tag{1}$$

It approximates the inner maximization by constructing adversarial examples based on natural examples, and then the model parameters $\theta$ are updated via an optimization method with respect to the adversarial examples, instead of the natural ones. One of the simplest choices of attack for adversarial training is the Fast Gradient Sign Method (FGSM) (Goodfellow et al., 2014):

$$x' = x + \epsilon \text{sign}(\nabla_x \mathcal{L}(f_\theta(x), y)) \tag{2}$$

Before the introduction of fast adversarial training (Wong et al., 2019), which we will introduce later, it was commonly believed that FGSM adversarial training fails to provide strong robustness (Kurakin et al., 2016). During FGSM adversarial training, the robust accuracy of the model would suddenly drop to almost 0% after a certain point, when evaluated against PGD attacks. This phenomenon was referred to as "catastrophic overfitting" in (Wong et al., 2019). The cause of catastrophic overfitting was studied extensively in (Tramèr et al., 2017): during training, since FGSM is a simple attack, the model learns to fool the FGSM attacks by inducing gradient masking/obfuscated gradient (Athalye et al., 2018); that is, the gradient is no longer a useful direction for constructing adversarial examples. The existence of catastrophic overfitting has prohibited the use of FGSM adversarial training.

To mitigate this issue, (Madry et al., 2017) introduced a multi-step variant of FGSM, namely Projected Gradient Descent (PGD), which takes multiple small steps with stepsize $\alpha$ to construct adversarial examples instead of one large step as in FGSM:

$$x'_{t+1} = \Pi_{\|x'-x\|_\infty \leq \epsilon} \left( x'_t + \alpha \text{sign}(\nabla_{x'_t} \mathcal{L}(f_\theta(x'_t), y)) \right) \tag{3}$$

Extensive experimental results (Madry et al., 2017; Athalye et al., 2018) have shown that, unless the model is particularly designed for creating obfuscated gradients (Tramer et al., 2020), PGD attacks are generally exempt from overfitting. Consequently, adversarial training with PGD leads to robust models against strong attacks, although its computational cost is often an order of magnitude more expensive than standard training and FGSM adversarial training.

Recently, in contrast to conventional believe, (Wong et al., 2019) proposed fast adversarial training and suggested it is possible to construct strongly robust models via FGSM adversarial training. They showed it is important to initialize a FGSM attack with large randomized perturbations, to protect FGSM adversarial training from overfitting. Although randomly initialized FGSM (R+FGSM) has been used in previous works (Tramèr et al., 2017), (Wong et al., 2019) points out that the scale of the randomized initialization was restrictive and needs to be enlarged. As a result, this simple modification enables R+FGSM adversarial training to obtain reasonable robustness against strong attacks.

### 3.2 SUB-OPTIMAL PERFORMANCE OF FAST ADVERSARIAL TRAINING

In (Wong et al., 2019) it is claimed that fast adversarial training has comparable performance as PGD adversarial training, yet they only compared to the original results from (Madry et al., 2017). Recent work (Rice et al., 2020) has shown PGD adversarial training can be greatly improved with the standard piecewise learning rate schedule and early stopping.

In Figure 1 we compare fast adversarial training (solid lines) and PGD adversarial training (dash lines) on a PreAct ResNet-18 (He et al., 2016a) model for classifying CIFAR-10 (Krizhevsky et al., 2009) under 10-step PGD attacks ($\epsilon = 8/255$). For fast adversarial training, we use a cyclic learning

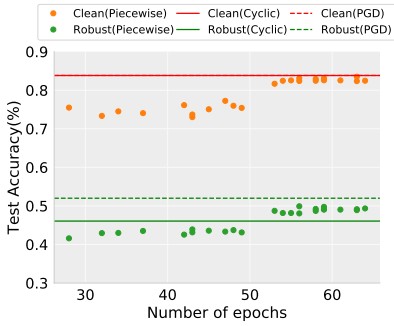

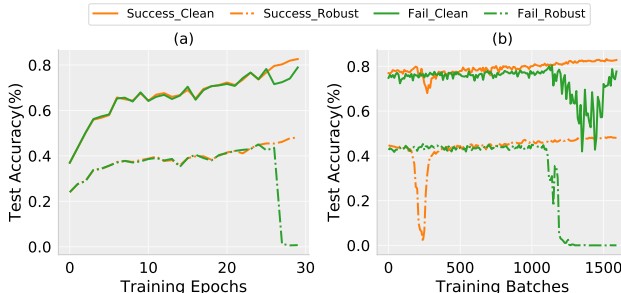

Figure 1: The epoch number where overfitting happens for FastAdv with the piecewise learning rate schedule, versus the best clean (orange) and robust (green) accuracy before overfitting. The solid lines and dashed lines are the clean and robust accuracy for FastAdv with cyclic learning rates and PGD adversarial training with piecewise learning rates.

Figure 2: Comparison of a success mode (orange) and a failure mode (green) of fast adversarial training for 30 epochs. In plot (a), we show the test clean and robust accuracy for each epoch. In plot (b), we report the same quantities for every 10 batches for the last 5 epochs. Both models encounter catastrophic overfitting, but only the model in the failure mode never recovers.

rate schedule (Smith & Topin, 2019), which linearly increases and decreases the learning rate. In particular, we linearly increase the learning rate from 0 to 0.2 in the first 12 epochs and decrease to 0 for the last 18 epochs as recommended in (Wong et al., 2019). For PGD adversarial training, we use a piecewise learning rate schedule which starts at 0.1 and decay by a factor of 0.1 at the 50th and the 75th epoch for a total of 100 epochs.

A clear gap in the robust accuracy is illustrated in Figure 1. There are two main factors accounting for this performance gap. First, although a model trained with the cyclic learning rate can converge in only a few epochs, it often results in sub-optimal results in the adversarial training setting (Rice et al., 2020) compared to the piecewise learning rate schedule. The issue is that fast adversarial training is forced to use a cyclic learning rate schedule. If a piecewise learning rate schedule is used for fast adversarial training for a large number of epochs, the model will still encounter catastrophic overfitting. We ran fast adversarial training with 25 different random seeds for 100 epochs, with the same piecewise learning rate schedule for PGD adversarial training, and terminated it when catastrophic overfitting happened. We add in Figure 1 the epoch number where the overfitting happens, versus the best clean and robust accuracy before overfitting.

The results show that none of the training progress exceeds even the 70th epochs without encountering catastrophic overfitting. For training progress terminated before the 50th epoch, where the learning rate drops, their performance is inferior due to insufficient training. On the other hand, although the rest of training progress also terminate early, they consistently outperformed fast adversarial training with the cyclic learning rate schedule. In other words, if fast adversarial training can run for more epochs with the piecewise learning rate schedule, it has the potential to improve upon fast adversarial training with the cyclic learning rate schedule.

Another reason for the inferior performance of fast adversarial training is the inherent weakness of FGSM compared to PGD attacks. As PGD is in general a better approximation to the solution for the inner maximization problem in (1), it is expected to produce more robust models. We seek to address this issue in Section 5.

### 3.3 Understanding Fast Adversarial Training

Although (Wong et al., 2019) has shown that initialization with a large randomized perturbation results in effective FGSM adversarial training, the underlying mechanism for its effectiveness remains a puzzle. Moreover, even with the recommended setting, catastrophic overfitting still happens on occasion. Plot (a) in Figure 2 shows both a success mode (orange) and a failure mode (green) when we use fast adversarial training with cyclic learning rate for 30 epochs. It seems that the model

in the success mode never encounters overfitting, while the model in the failure mode encounters catastrophic overfitting at around the 26th epoch. However, surprisingly, if we look closer at the training progress in plot (b), where we report the test clean and robust accuracy for every 10 batches (with a batch size of 128) for the last 5 epochs, it is observed that the model in the success mode also encounters a sudden drop in test robust accuracy, indicating catastrophic overfitting, but it recovers immediately.

This finding explains why fast adversarial training could run more epochs than vanilla FGSM adversarial training. It is not because it can completely avoid catastrophic overfitting, as previously believed; rather, it is because it can recover from the catastrophic overfitting in a few batches. This has not been observed before because normally a model is only evaluated per epoch, while such "overfit-and-recover" behavior happens within a span of a few batches.

The observation in Figure 2 also suggests that, when catastrophic overfitting happens, although the model quickly transforms into a non-robust one, it is fundamentally different from an ordinary non-robust model. In fact, the non-robust model due to catastrophic overfitting can quickly retain its robustness once the corresponding attack find the correct direction for constructing attacks again. Thus, the root of the effectiveness of randomized initialization in fast adversarial training is its ability to help escaping from catastrophic overfitting. On the other hand, it also explains why fast adversarial training still overfits after long training progress. Randomized initialization works with a high probability, but not always. As the training progress continues, the model is more capable of overfitting, and fast adversarial training is less likely to find the correct direction for constructing FGSM attacks.

To verify our analysis, we conduct experiments to exclude the use of randomized initialization, that is to use vanilla FGSM adversarial training, but also run PGD adversarial training for a few batches when catastrophic overfitting happens. In particular, we monitor the PGD robust accuracy on a validation set during training. Once there is a sudden drop of the validation robust accuracy, which indicates the occurrence of overfitting, we run PGD adversarial training for a few batches to help the model recover from overfitting, as an alternative to R+FGSM. The same piecewise learning rate schedule as in Figure 1 is used. We also run the vanilla fast adversarial training as a reference.

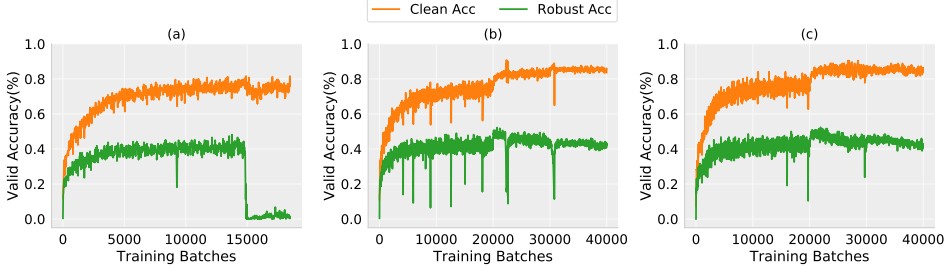

Figure 3: Validation clean and robust accuracy evaluated for every 20 batches. (a) Fast adversarial training. (b) FGSM with no randomized initialization but using PGD to mitigate overfitting. (c) FGSM with randomized initialization and PGD to mitigate overfitting. All training uses the same piecewise learning rate schedule.

The results are illustrated in Figure 3, where we report the robust accuracy under 10-step PGD attacks with size $\epsilon = 8/255$ for every 20 batches. Comparing plot (a) and (b), while the fast adversarial training overfits after around 15,000 batches, FGSM adversarial training without randomized initialization obtains robust accuracy despite several "overfit-and-recover" behaviors. This confirms our hypothesis that the essential nature of successful FGSM adversarial training is not the randomized initialization, but the ability to recover from catastrophic overfitting.

## 4    A SIMPLE FIX TO FAST ADVERSARIAL TRAINING

The analysis and experimental results above suggest that ($i$) FGSM adversarial training is useful as long as it can recover from catastrophic overfitting, and ($ii$) fast adversarial training can only run for limited epochs because the randomized initialization is not reliable. Therefore, a more reliable way

to mitigate catastrophic overfitting is needed. To this end, we propose a simple fix to fast adversarial training, incorporating PGD adversarial training when catastrophic overfitting is observed.

---

**Algorithm 1** Improved fast adversarial training for $T$ epochs, given some radius $\epsilon$, $N$ PGD steps, step size $\alpha$, a threshold $c$, frequency of detection $s$, and a dataset of size $M$ for a network $f_\theta$.

---

**for** $t = 1 \ldots T$ **do**
    **for** $i = 1 \ldots M$ **do**
        **if** $\text{Acc}_{\text{last}} > \text{Acc}_{\text{valid}} + c$ **then**
            $\delta = \text{PGD Attack}(f_\theta, x_i, y_i)$  *// Overfitting happens, run PGD adversarial training*
        **else**
            $\delta = \text{R+FGSM Attack}(f_\theta, x_i, y_i)$  *// No Overfitting, use R+FGSM adversarial training*
        **end if**
        $\theta = \theta - \nabla_\theta \ell(f_\theta(x_i + \delta), y_i)$ *// Update model weights with some optimizer, e.g. SGD*
    **end for**
    **if** i%s == 0 **then**
        Let $\text{Acc}_{\text{last}} = \text{Acc}_{\text{valid}}$. Update robust accuracy $\text{Acc}_{\text{valid}}$ under PGD attacks.
    **end if**
**end for**

---

The proposed approach is described in Algorithm 1. The core idea is simple and has been described briefly in the previous section: we hold out a validation set and monitor its robust accuracy for detecting overfitting. When there is a drop on the validation robust accuracy beyond a threshold, at which point catastrophic overfitting happens, we run 10-step PGD adversarial training for a few batches to help the model regain its robustness.

Note that although the training progress in plot (a) of Figure 3 overfits at around the 15,000th batch, the frequency of catastrophic overfitting is much lower compared to the training progress in plot (b), where no randomized initialization is used. This also imply that although randomized initialization cannot prevent catastrophic overfitting, it effectively reduces its occurrences. To verify this conjecture, we perform the same experiment as in plot (b), but now with randomized initialization, and show the training progress in plot (c) of Figure 3. The occurrences of catastrophic overfitting is much fewer than in plot (b), confirming our conjecture. Therefore, we keep the large randomized initialization for FGSM adversarial training in Algorithm 1, resulting in R+FGSM adversarial training. The infrequent occurrences of catastrophic overfitting also ensures the additional PGD adversarial training adds little computational overhead to the training progress.

Table 1: CIFAR-10 standard and robust accuracy on PreAct ResNet-18 for vanilla PGD adversarial training with early stopping, fast adversarial training (FastAdv), improved fast adversarial training (FastAdv+) and fast adversarial training as a warmup for PGD adversarial training (FastAdvW).

| Method | Clean Accuracy | PGD($\epsilon = 8/255$) | Time/Epoch(min) | Total Time(min) |
|--------|---------------|------------------------|-----------------|-----------------|
| PGD | $83.43 \pm 0.25\%$ | $51.74 \pm 0.17\%$ | 1.66 | 166.45 |
| FastAdv | $83.41 \pm 0.13\%$ | $46.14 \pm 0.08\%$ | 0.31 | 12.35 |
| FastAdv+ | $\mathbf{83.54 \pm 0.22}\%$ | $48.43 \pm 0.14\%$ | 0.32 | 32.14 |
| FastAdvW | $83.18 \pm 0.18\%$ | $\mathbf{53.09 \pm 0.11}\%$ | 0.41 | 40.73 |

**Hyperparameters** In Table 1 we report the final clean and robust accuracy of the improved fast adversarial training (FastAdv+). We use the same piecewise learning rate schedule and early stopping as used in PGD adversarial training. We detect overfitting every $s = 20$ batches with a randomly sampled validation batch, and PGD adversarial training runs for $s = 20$ batches until the when overfitting happens. The threshold for detecting catastrophic overfitting is $c = 0.1$. The robust accuracy is evaluated against 50-step PGD attacks with 10 restarts for $\epsilon = 8/255$. Note we use half-precision computations (Micikevicius et al., 2017) as recommended in (Wong et al., 2019) for all training methods, for acceleration. All experiments are repeated for 5 times, and both the mean and the standard deviation are reported.

**Efficiency** Although FastAdv consumes less time, thus seems to be more efficient, it is worth noting that the computational time per epoch is almost the same for FastAdv and FastAdv+. FastAdv consumes less time merely because the training progress is forced to stop before its convergence. On

the other hand, our proposed FastAdv+ allows the training process to converge and results in better performance.

## 5 Fast Adversarial Training as a Warmup

We are able to improve the performance of fast adversarial training by allowing longer training progress. However, the associated robust accuracy is still noticeably worse than PGD adversarial training. This is expected, as PGD is inherently a stronger attack than FGSM. In this section, we adapt a previously studied technique (Wang et al., 2019), using FGSM adversarial training as a warmup for PGD adversarial training, to close the gap between fast adversarial training and PGD adversarial training.

It has been observed in (Wang et al., 2019) that using FGSM at the early stage of PGD adversarial training does not degrade its performance, and even provides slight improvement. The intuition behind this is that at the early stage of training, the model is vulnerable to adversarial attacks, and therefore there is no difference between using a weak attack and a strong attack. As the training proceeds, the model becomes more robust to weak attacks, and sometimes even overfits, where stronger attacks are more effective at increasing the robustness of the model.

However, due to the risk of catastrophic overfitting, only a few epochs of FGSM adversarial training were used in (Wang et al., 2019) as a warmup, and consequently it does not provide much improvement on the robust accuracy, nor does it save much training time. As FastAdv+ can run for as many epochs as needed, it is possible to use it for most of the training epochs and PGD adversarial training for only a few epochs at the end.

**Starting Point of PGD Adversarial Training**   Since early stopping always happens a few epochs after the learning rate decay, we starts PGD adversarial training a few epochs before the learning rate decay, to minimize the span of PGD adversarial training for the purpose of efficiency. In the experiments, we run the improved fast adversarial training for the first 70 epochs and change to PGD adversarial training. The early stopping happens at the 78th epoch, meaning we only run PGD adversarial training for no more than 10 epochs.

We report in Figure 4 the validated clean and robust accuracy during the whole training progress for FastAdv+ as a warmup, termed FastAdvW. While using FastAdvW improves upon FastAdv+, it still suffers from overfitting (however, *not* catastrophic overfitting) in the later stage of training. We assume this is due to the fact that the FGSM attack with a large randomized initialization is already strong, in contrast to vanilla FGSM adversarial training used in (Wang et al., 2019). Following the intuition that only a weak attack is needed in the early stage, we reduce the size of perturbation $\epsilon$ from $8/255$ to $4/255$ for the stage of fast adversarial training. As shown in Figure 4, this change of the attack size (FastAdvW 4-8) allows the model to reach higher robust accuracy. We also report the final test clean and robust accuracy, which is based on early

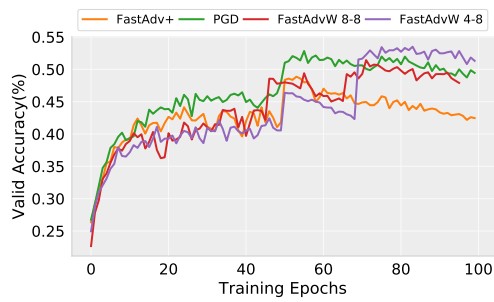

Figure 4: Robust accuracy on a validation set, evaluated for FastAdv+, PGD adversarial training, FastAdvW with a constant perturbation size $\epsilon = 8/255$, and with $\epsilon = 4/255$ and $8/255$ in two stages.

stopping[1], on the test set in Table 1. This shows FastAdvW outperforms PGD adversarial training in robust accuracy and is comparable on clean accuracy, while consuming much less time.

## 6 Additional Experiments

In the above analyses, we only evaluated the proposed approach on CIFAR-10 using the PreAct ResNet-18 architecture. In this section, we run more experiments on various data sets and model architectures to show the generality of our results.

---

[1]For FastAdvW, we only consider stopping after PGD adversarial training starts.

We first show in Table 2 results for both CIFAR-10 and CIFAR-100 on the Wide-ResNet 34-10 (Zagoruyko & Komodakis, 2016) as this model structure is widely used in the adversarial training literature (Madry et al., 2017; Shafahi et al., 2019; Zhang et al., 2019b; Rice et al., 2020). The same setting of hyperparameters in Section 4 is used, except the threshold for detecting "catastrophic overfitting" is reduced to $c = 0.05$ for CIFAR-100 to accommodate for its range of robust accuracy.

In addition, we also include in Table 2 results for "free" adversarial training (Shafahi et al., 2019). This approach reduces the computational cost of PGD adversarial training utilizing the "minibatch replay" technique, which adds adversarial perturbations and updates the model simultaneously on the same minibatch for several iterations, to imitate PGD adversarial training. As a result, this approach only needs to run for several epochs to converge. In this experiments, we follow the recommendation in (Shafahi et al., 2019) and replay each batch $m = 8$ times for a total of 25 epochs.

Finally, we conduct experiments on Tiny ImageNet, with results also summarized in Table 2. Although previous works (Wong et al., 2019; Shafahi et al., 2019) conduct experiments on ImageNet, it still requires several GPUs to run. As we only run experiments on a single RTX 2080ti, we considered Tiny ImageNet, which consists of 200 ImageNet classes at 64 x 64 resolution. The architecture we use is ResNet-50 (He et al., 2016b) and the hyperparameters, such as learning rate and size of attacks, are kept the same as for CIFAR datasets.

Table 2: CIFAR-10 and CIFAR-100 standard and robust accuracy, and the corresponding training time on Wide-ResNet 34-10 for PGD adversarial training, free adversarial training (Free), FastAdv, FastAdv+ and FastAdvW. PGD attacks with 50 iterations and 10 restarts are used for evaluation.

| Data | Method | Clean Accuracy | PGD($\epsilon = 8/255$) | Total Time(min) |
|---|---|---|---|---|
| CIFAR-10 | PGD | $85.27 \pm 0.31\%$ | $54.10 \pm 0.20\%$ | 998.27 |
| | Free (m=8) | $85.87 \pm 0.37\%$ | $46.13 \pm 0.19\%$ | 358.53 |
| | FastAdv | $85.21 \pm 0.22\%$ | $46.36 \pm 0.13\%$ | 82.32 |
| | FastAdv+ | $\mathbf{86.52 \pm 0.25}\%$ | $51.01 \pm 0.18\%$ | 143.17 |
| | FastAdvW | $85.91 \pm 0.36\%$ | $\mathbf{55.13 \pm 0.22}\%$ | 237.87 |
| CIFAR-100 | PGD | $61.92 \pm 0.20\%$ | $26.60 \pm 0.14\%$ | 1014.47 |
| | Free (m=8) | $\mathbf{62.11 \pm 0.22}\%$ | $25.37 \pm 0.09\%$ | 362.83 |
| | FastAdv | $55.71 \pm 0.17\%$ | $27.50 \pm 0.13\%$ | 103.83 |
| | FastAdv+ | $60.57 \pm 0.25\%$ | $28.61 \pm 0.10\%$ | 184.50 |
| | FastAdvW | $61.01 \pm 0.27\%$ | $\mathbf{28.88 \pm 0.17}\%$ | 257.13 |
| Tiny ImageNet | PGD | $45.34 \pm 0.30\%$ | $21.62 \pm 0.11\%$ | 2099.83 |
| | Free (m=8) | $34.75 \pm 0.11\%$ | $14.30 \pm 0.07\%$ | 743.94 |
| | FastAdv | $\mathbf{48.31 \pm 0.21}\%$ | $19.96 \pm 0.08\%$ | 194.51 |
| | FastAdv+ | $48.02 \pm 0.19\%$ | $20.05 \pm 0.11\%$ | 330.65 |
| | FastAdvW | $46.73 \pm 0.31\%$ | $\mathbf{21.82 \pm 0.19}\%$ | 592.22 |

The results in Table 2 are consistent with what we have observed on the PreAct ResNet-18 architecture for CIFAR-10. While FastAdv+ outperforms vanilla fast adversarial training as a result of longer training progress, its robust accuracy is no better than PGD adversarial training. However, when we use FastAdv+ as a warmup, its clean accuracy becomes comparable to PGD adversarial training, while its robust accuracy constantly outperforms PGD adversarial training. In addition, FastAdvW only consumes 25% of the training time of PGD adversarial training. It is also worth noting that although free adversarial training uses a piecewise learning rate schedule as well, it only obtains its best performance at the end of the training progress, thus does not benefit from early stopping in both the performance and the efficiency.

## 6.1 SANITY CHECK

We now perform sanity check following the suggestions from (Athalye et al., 2018) to ensure our proposed approaches are truly robust. For all the sanity checks, we evaluate our approaches on both PreAct ResNet-18 and Wide-ResNet 34-10 for classifying CIFAR-10.

We first show the clean and robust accuracy under 10-step PGD attacks under varying sizes from 0 to $12/255$ in Figure 5. It can be observed that the decreasing trend of robust accuracy of our proposed approaches are consistent with models trained via PGD-AT and FastAdv, where FastAdvW outperforms PGD-AT and FastAdv+ outperforms FastAdv consistently.

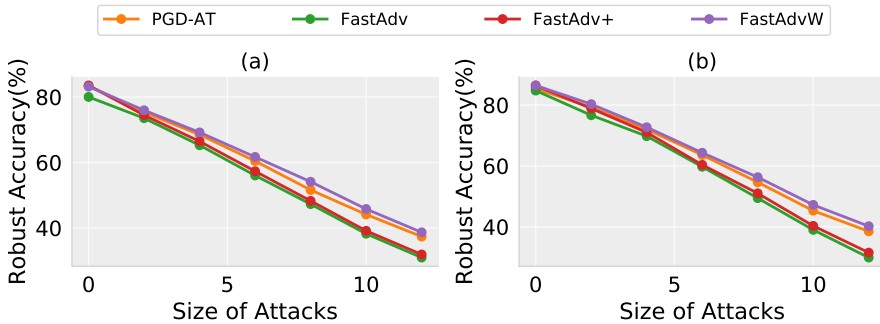

Figure 5: Robust accuracy under 10-step PGD with step sizes from 0 to 12/255. The evaluations are performed on (a) PreAct ResNet-18 and (b) Wide-ResNet 34-10 for classifying CIFAR-10. We also include results for PGD-AT, FastAdv for comparison.

In addition, we show robust accuracy under other types attacks that do not rely on estimating gradients. AutoAttack (Croce & Hein, 2020) ensembles four diverse attacks to reliably evaluate robustness: two improved PGD attacks, a boundary based attack FAB (Croce & Hein, 2019), and a query-based black-box attack Square Attack (Andriushchenko et al., 2019). We also use the recently proposed state-of-the-art black-box attacks (Tashiro et al., 2020) to evaluate our models. This attack utilizes output diversified sampling to construct transferable black-box attacks and is almost as powerful as white-box PGD attack. The results are summarized on Table 3.

Table 3: CIFAR-10 standard and robust accuracy under various types of attacks on PreAct ResNet-18 and Wide-ResNet 34-10 for PGD adversarial training, FastAdv, FastAdv+ and FastAdvW.

| Architecture | Method | Clean Accuracy | PGD-10 | AutoAttack | Transfer |
|---|---|---|---|---|---|
| ResNet-18 | PGD | 83.21% | 51.62% | 46.75% | 56.13% |
| | FastAdv | 83.55% | 46.86% | 41.63% | 49.86% |
| | FastAdv+ | **83.59**% | 48.29% | 43.40% | 52.06% |
| | FastAdvW | 83.11% | **54.09**% | **47.61**% | **57.13**% |
| WideResNet | PGD | 85.42% | 54.70% | 49.91% | 59.94% |
| | FastAdv | 85.34% | 48.55% | 43.58% | 51.98% |
| | FastAdv+ | 86.43% | 50.11% | 44.83% | 55.39% |
| | FastAdvW | **86.44**% | **55.50**% | **50.57**% | **62.16**% |

We observe that no attack significantly reduces the robust accuracy of our proposed approaches, and the relative performance across different approaches are consistent. AutoAttack reduces approximately 5% more robust accuracy than 10-step PGD, which is consistent with the observations in (Croce & Hein, 2020).

## 7 CONCLUSION

We have conducted experiments to show that the key to the success of FGSM adversarial training is the ability to recover from "catastrophic overfitting". Fast adversarial training utilizes randomized initialization to achieve this goal but still suffers from catastrophic overfitting for a large number of training epochs. We design a new training strategy that mitigates this caveat and enables the commonly used piecewise learning rate schedule for fast adversarial training and, as a result, improves clean and robust accuracy. We also use the improved fast adversarial training as a warmup for PGD adversarial training, and find it is sufficient to use this warmup for a majority of the training epochs to save time and further improve model robustness. As a result, we obtain superior performance to the expensive, state-of-the-art PGD adversarial training with much-reduced training time. Our proposed approaches are easy to implemented, thus could be used as baselines for empirical adversarial defense.

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

## A  APPENDIX

## B  STRONG ATTACKS AFTER OVERFITTING

When FastAdv+ is used for training a model, even though the model can recover from catastrophic overfitting via PGD adversarial training, it is possible that the model overfits to PGD attacks and stays vulnerable to other attacks. Therefore, we extract the model right after its recovery from catastrophic overfitting and run several kinds of attacks, including 10-step PGD attacks, 50-step PGD attacks with 10 restarts, C&W attacks (Carlini & Wagner, 2017) and fast adaptive boundary (FAB) attacks (Croce & Hein, 2019), on this model.

Table 4:  CIFAR-10 standard and robust accuracy on PreAct ResNet-18 under various types of attacks.

| Attacks | PGD-10 | PGD-50 | C&W | FAB |
|---------|--------|--------|------|------|
| Robust Accuracy | 40.22% | 39.41% | 41.05% | 38.68% |

The result shows the model recovered from catastrophic overfitting is indeed robust. Note the robust accuracy is relatively low as we are not using the final model.

## C  ABLATION ANALYSIS ON ADJUSTED ATTACK SIZE

In Section 5, we show it is possible to improve the performance of FastAdvW via using a smaller size of attacks for FGSM adversarial training. It is possible that the adjusted size of attacks benefits not only our approach, but also PGD adversarial training. Therefore, we use the same setting ($4/255$ for the first 70 epochs and $8/255$ for the rest) for full PGD adversarial training and compare it to vanilla PGD adversarial training.

Table 5:  CIFAR-10 standard and robust accuracy on PreAct ResNet-18 for vanilla PGD adversarial training and PGD adversarial training with adjusted size of attacks ($4/255$ and $8/255$).

| Method | Standard Accuracy | PGD($\epsilon = 8/255$) |
|--------|-------------------|--------------------------|
| PGD | $83.43 \pm 0.25\%$ | $51.74 \pm 0.17\%$ |
| PGD(adjusted size) | $83.11 \pm 0.11\%$ | $52.14 \pm 0.28\%$ |

The results show that PGD adversarial training enjoys limited benefits from the adjusted size of attacks. This strategy is more compatible with our proposed FastAdvW.

