# OpenReview forum: "Towards Understanding Fast Adversarial Training"
_ICLR.cc/2021/Conference — Reject_

### Official Review · AnonReviewer4 · 2020-10-27
**A good paper but needs improvements**

**Rating:** 5
**Confidence:** 5

**Review:**

This paper proposes an improvement over fast adversarial training to improve the robustness of the model in an efficient manner. Overall it's a well written paper but it can be improved in certain ways as follows...

- Fig. 3 lacks important information about what specific attack was used to compute the robust accuracy. was it PGD? If yes, what are the PGD parameters?
- It's not clear if the improvements are due to few iterations of PGD or due to piecewise linear learning rate regime? An explicit experiments comparing the two learning rate regimes (cyclic and piecewise) would be good to confirm.
- For section 5, when authors are using FGSM as a warm-up, they should refer/compare with https://arxiv.org/pdf/2002.04237.pdf where clean training is used as warm-up. Author mention that strength of the adversary doesn't matter in the initial phase of training, so should compare results with varying strength going to zero.
- In most of the experiments, the PGD step size is missing. Moreover, it will be good to see the robustness of this technique against varying PGD step size.
- I think authors use R+FGSM and fast adversarial training interchangeably which creates some confusion.. it will be nice to be consistent with the terminology.
-

---

> ### Author Response · Authors · 2020-11-14
> **Thank you for the valuable comments**
>
> Thank you for the valuable comments. We now respond to individual comments in the following:
>
> 1) We used PGD-10 to evaluate the robust accuracy in Figure 3 as FGSM is not sufficient to discover catastrophic overfitting. The hyper-parameters are the same as the ones used for PGD adversarial training, which is 10 steps with step size 2/255.
>
> 2) Please note PGD adversarial training only happens in a few batches that do not sum to even one epoch, thus is unlikely to be the reason for the improvement. We used the cyclic learning rate schedule for FastAdv+ on ResNet18 for classifying CIFAR10 and found the clean accuracy (83.31%) and robust accuracy (46.33%), which is close to the original FastAdv (83.41%/46.14%) Therefore, we believe the improvement is due to the learning rate schedule.
>
> 3) Thank you for pointing out related work. We find their results on WideResNet are close to ours but are much worse (71.0%/40.6% vs 83.18%/53.19%) on ResNet-18. Our choice of 4/255 at the initial phase is determined via experiments over varying strength, and we find 4/255 performs the best on all the architectures and data sets. We will include more comprehensive comparisons in the next revision.
>
> 4) Unless specified, we use 10-step PGD with 2/255 step size and 8/255 attack size for adversarial training. We include results for various sizes of attacks in the revised version.
>
> 5) We apologize for the confusion. We intend to use fast adversarial training as a summarization of the specific setting in Wong et al. 2020, including R+FGSM adversarial training, a large random initialization U(-8/255, 8/255), and the cyclic learning rate schedule.

---

### Official Review · AnonReviewer2 · 2020-10-28
**Interesting paper, needs more evaluation**

**Rating:** 7
**Confidence:** 5

**Review:**

################################## Summary ##################################

This paper shows that the main reason for the success of Fast Adversarial Training ([1], will be referred to as FBF in this review) is its ability to recover from catastrophic overfitting. Based on this observation, the authors propose to utilize PGD multi-step training for a few iterations (when catastrophic overfitting occurs), and resume single-step training after the model recovers. Further, the authors also propose to use this improved Fast Adversarial Training (FastAdv+) as a warmup for PGD Adversarial Training, and demonstrate improved performance over PGD-AT, at a significantly lower computational cost.

[1] Wong et al. Fast is Better than Free: Revisiting Adversarial Training, ICLR 2020

################################### Pros #####################################

  -  The authors present a very interesting finding, that FBF has catastrophic overfitting for a few intermediate iterations, and recovers very quickly from this.
  -  Based on this observation, they propose to use PGD based training for a few intermediate iterations, which seems to prevent this overfitting effectively
  -  The authors also propose the use of this training as warm up for PGD training, and demonstrate significantly improved results.
  -  This is certainly a significant contribution of the paper since it achieves improved results at a much lower computational cost.

################################### Cons #####################################

  -  Since the proposed defenses involve single-step training, the absence of gradient masking needs to be justified using thorough validation as discussed by Carlini et al. [1], using gradient-free attacks such as Square attack/ SPSA, black-box transfer based attacks, attacks with multiple steps and multiple random restarts. Also, the sanity checks proposed by Athalye et al. [2] need to be demonstrated.
  -  The paper discusses results only on PGD attack, which is not the current state-of-the-art. The proposed defenses (FastAdv+, FastAdvW) must be evaluated on stronger attacks such as AutoAttack [3] and MultiTargeted Attack [4].
  -  While the use of the modified FGSM attack would improve the efficiency of PGD training, it is not clear why it should lead to improved robustness.
  -  Could the authors clarify if all the other training/ hyperparameter settings (such as batch size, weight decay, optimizer, initial learning rate and schedule, number of epochs, initial random noise added for the attack, validation split, use of early stopping, batch norm in train/ eval mode during attack generation [5]) are similar for PGD-AT (reported in Table-4 of the Appendix) and FastAdvW (reported in Table-1)?
  -  The learning rate schedule used for PGD-AT (Sec. 3.2) is different from that used by Rice et al. So, the results could be sub-optimal. Could the authors use similar settings as Rice et al. (SGD optimizer using a batch size of 128, a step-wise learning rate decay set initially at 0.1 and divided by 10 at epochs 100 and 150, total epochs 200 and weight decay 5e-4) for reporting the PGD baseline results?

[1] Carlini et al., On evaluating Adversarial Robustness, https://arxiv.org/abs/1902.06705

[2] Athalye et al., Obfuscated Gradients Give a False Sense of Security: Circumventing Defenses to Adversarial Examples, ICML 2018

[3] Croce et al., Reliable Evaluation of Adversarial Robustness with an Ensemble of Diverse Parameter-free Attacks, ICML 2020

[4] Gowal et al., An Alternative Surrogate Loss for PGD-based Adversarial Testing, https://arxiv.org/pdf/1910.09338.pdf

[5] Bag of Tricks for Adversarial Training, https://openreview.net/forum?id=Xb8xvrtB8Ce

############################## Reasons for score #################################

The paper highlights a very interesting finding in FBF training, proposes a single-step defense, and also an improvement to speed-up PGD-AT. However, it does not show sufficient experimental results for reliable evaluation of the defenses and to ensure the absence of gradient masking. Hence I think the paper is marginally below the acceptance threshold. I will be happy to increase the score if the required experimental results are presented during the rebuttal.

######################### Questions during rebuttal period ###########################

  -  Could the authors provide the following results (for CIFAR-10) for both the proposed defenses (FastAdv+, FastAdvW).
        *  Evaluation against AutoAttack [3] and MultiTargeted Attack [4]. (It would help to also report the corresponding results for the baselines - FBF, PGD-AT)
        *  Plot of robust accuracy vs. attack distortion bound (epsilon for PGD-10 step attack) as discussed by Athalye et al. [2]
        *  Accuracy on black box transfer based attacks (using normally trained model of the same architecture as a source)
  -  Could the authors also report PGD-10 step accuracy for PGD-AT and FastAdvW for CIFAR-10 dataset on PreActResNet-18 and WRN-34-10? This would help with comparison of baselines against those reported in prior work.
  -  In the proposed defense FastAdv+, could the authors clarify what is the fraction of validation split used for detecting catastrophic overfitting and what is the number of steps used for the PGD attack on the validation set? Is the validation time included in the time reported in Table-1? If not, could the authors report the total time including validation?
  -  Could the authors clarify the size of the validation split used for early stopping (also in Fig.4) and the number of steps used for the PGD attack for early stopping? Is this consistent across all experiments?
  -  (good to have) Loss surface plots for the proposed defenses to show the absence of gradient masking. (similar to those reported in [5])

################## Additional Feedback (not part of decision assessment) ###################

  -  The authors mention the following: “although the model quickly transforms into a non-robust one, it is fundamentally different from an ordinary non-robust model”. It would be insightful to study the properties of this intermediate model that suffers from catastrophic overfitting, to understand more about why it is able to recover so quickly.
  -  The authors could visualize the loss surface of the FBF trained model during the catastrophic overfitting, and immediately after it recovers. This can lead to insightful findings on what is happening in the vicinity of the data samples in very few iterations. The loss surface can be plotted similar to Fig.1 in [5].

[5] Tramer et al., Ensemble Adversarial Training: Attacks and Defenses, ICLR 2018

############################# Update after rebuttal ###############################

The authors response addresses my concerns and I would like to update my score to 7. The paper presents insightful findings about FBF, and proposes a simple and effective method for stabilizing single-step adversarial training. This is useful not only for FBF but also for other single-step defenses. Although the gain in robustness is marginal, stabilizing single-step training is useful. The authors also propose a computationally efficient method of achieving robustness similar to PGD training.

---

> ### Author Response · Authors · 2020-11-14
> **Thank you for the valuable comments**
>
> Thank you for the valuable comments. We now respond to individual comments in the following:
>
> 1) We now have included results for AutoAttack, which consists of Square Attack, a boundary-based attack FAB and two improved PGD attacks. We also report results for black-box transfer attacks, and PGD attacks with various sizes in the revised paper. We believe it is sufficient to ensure there is no gradient masking.
>
> 2) The reason FGSM adversarial training as warmup improved PGD adversarial training is based on the conclusion from [1] where the authors found “better robustness of adversarial training is associated with training on high quality adversarial examples in the later stages. In the early stages, high convergence quality adversarial examples are not necessary and can even be harmful.” Thus, we try to use FastAdv+ in the initial stage as much as possible.
>
> 3) Our hyperparameters setting follows the ones from [2], unless specified in section 4. The learning rate schedule is also the same as the ones from [2], where it is divided by 10 at 50% and 75% steps, but we only run 100 epochs instead of 200 due to computational limitations (especially as we repeat our experiments multiple times). When using 200 epochs, PGD-AT on ResNet-18 improves from 83.65%/51.74% to 83.29%/52.17%, while FastAdvW improves from 83.18%/53.09% to 83.03%/53.59%. The relative performance is almost the same.
>
> 4) We originally reported results for PGD-50 with 10 random starts. We include results for PGD-10 in the added table for ease of comparison to other baselines.
>
> 5) We use 2.5% data from the training set as the validation set. This validation set is used for both monitoring the occurrence of catastrophic overfitting and early stopping. We include the validating time as a part of the training time.
>
> 6) We consistently use the robust accuracy under PGD-10 for early-stopping.
>
> 7) We appreciate the suggestions and will consider adding illustrations of the loss surfaces of models in the next revision.
>
> [1] Wang, Yisen, et al. "On the Convergence and Robustness of Adversarial Training." International Conference on Machine Learning. 2019.
>
> [2] Rice, Leslie, Eric Wong, and J. Zico Kolter. "Overfitting in adversarially robust deep learning." arXiv preprint arXiv:2002.11569 (2020).

---

> > ### Comment · AnonReviewer2 · 2020-11-18
> > **FastAdv results seem lower than expected**
> >
> > I would like to thank the authors for their detailed response and additional results. This addresses most of my concerns.
> >
> > The AutoAttack results on FastAdv in Table-3 seem lower than expected. In the AutoAttack github repo (https://github.com/fra31/auto-attack) the number reported is 43.21. If we consider this to be the baseline, improvement achieved is marginal for FastAdv+. Could the authors clarify the same?
> >
> > Secondly, PGD adversarial training using settings by Rice et al. achieves an AA accuracy of 52.6% for WRN-34-10. However, the reported accuracy in Table-3 is 49.9%. Could the authors clarify whether the difference is due to the use of 100 epoch schedule rather than 200-epoch schedule?

---

> > > ### Author Response · Authors · 2020-11-19
> > > **Thank you for the response!**
> > >
> > > Regarding the results for FastAdv, we use 20 epochs instead of 30 epochs as in the original paper. This is because although 30 epochs achieve better performance, there is a chance of encountering catastrophic overfitting (similar to what we observed in Figure 1), which undermines the results when we conduct repeated experiments. Note in the original paper, the authors claimed catastrophic overfitting does not exist when alpha=10/255 (appendix D), yet they only ran the experiment 3 times. In contrast, we observe catastrophic overfitting when we run the experiment more than 20 times. This issue is severer on WRN-34-10, where catastrophic overfitting could happen within the first 15 epochs, thus we ended up using 12 epochs, resulting in an even larger gap between FastAdv and FastAdv+. Even without considering the occurrence of catastrophic overfitting, the piecewise decay schedule outperforms the cyclic schedule in general according to Rice et al.
> > >
> > > There are two factors contributing to the gap of PGD adversarial training on WRN: 1) we use fewer epochs than in the original paper; 2) we used 2.5% training data as the validation set, instead of 2% in the original paper. We believe it is still a fair comparison as we used the same setting for both PGD and FastAdvW.

---

> > > > ### Comment · AnonReviewer2 · 2020-11-21
> > > > **FastAdv+ with cyclic schedule**
> > > >
> > > > I thank the authors for their response.
> > > >
> > > > If 30 epoch training fails for FBF cyclic schedule, the failure can be prevented by using the method proposed in this paper (using PGD adversaries for a few iterations). In such a case, the accuracy would be similar to that reported by FBF in the autoattack table (43.21%). In this case, the advantage with piecewise linear schedule is marginal. The same holds for WRN training as well.
> > > > I believe results of FastAdv+ with cyclic learning rate should also be included in Table-3. Also, in this experiment, the total number of epochs (20/30/40/more) is a hyperparameter that needs to be selected based on best results. This can be studied for both ResNet-18 and WRN.
> > > > Although piecewise schedule may give results similar to cyclic schedule, the proposed method is still useful in stabilizing the results of FBF. The observations about piecewise learning rate being better than cyclic learning rate in the work by Rice et al. are specific to PGD multi-step adversarial training, and this may be different for other cases.

---

> > > > > ### Author Response · Authors · 2020-11-22
> > > > > **We appreciate the valuable advice**
> > > > >
> > > > > We want to emphasize that the focus of our paper is to propose an approach to allowing more epochs in R+FGSM adversarial training, instead of the difference between two different learning rate schedules. We actually agree that the gap could be marginal if we could run more epochs with the cyclic schedule, but it cannot be done without our proposed approach. With our proposed approach, both the piecewise decay schedule and cyclic schedule should outperform the original FastAdv.
> > > > >
> > > > > If we care about the cyclic schedule, there could also be a cyclic schedule version of FastAdvW where we shift to PGD adversarial training at the last several epochs. Eventually, the difference between two learning rate schedules is an interesting topic but is orthogonal to the focus of our paper, which is stabilizing R+FGSM adversarial training. We use the piecewise decay schedule mainly because it performs well and is commonly used in the literature. On the other hand, FastAdv is forced to use the cyclic schedule because it can only run a limited number of epochs.
> > > > >
> > > > > In addition, we believe studying the best total number of epochs for the cyclic learning rate is also beyond the scope of this paper.

---

> > > > > > ### Comment · AnonReviewer2 · 2020-11-22
> > > > > > **Improving clarity of Section 3.2**
> > > > > >
> > > > > > While I agree that the primary contribution of the paper is to run R+FGSM training for longer, parts of the paper (specifically Section-3.2) indicate that use of longer step-wise learning schedule gives better results compared to cyclic schedule. In this case it would be confusing whether the benefit is due to longer training/ step-wise schedule/ both. It would be helpful to improve the clarity to indicate the importance of longer training alone.

---

> > > > > > > ### Author Response · Authors · 2020-11-22
> > > > > > > **Add to the next revision**
> > > > > > >
> > > > > > > Thank you for the quick response and suggestion. Our current experiment results suggest cyclic schedule indeed has a similar performance with piecewise decay when longer training is used for both.
> > > > > > >
> > > > > > > We agree adding the results for the cyclic schedule with longer training is a proper and important ablation study. We will add it to the next revision and also modify the text in Section 3 to make it clear.

---

### Official Review · AnonReviewer3 · 2020-10-29
**Improving PGD adversarial training with FGSM adversarial training as warmup (mitigate the overfitting issue with PGD)**

**Rating:** 5
**Confidence:** 4

**Review:**

Summary:
This paper proposes a method called: improved fast adversarial training (FastAdv+) which improves fast adversarial training by replacing randomized initialization with PGD adversarial training when there is overfitting issue during training.

Strengths:
The idea is reasonable and easy to implement. There is a marginal improvement of accuracy under PGD attack ($\epsilon=8/255$) of around 2%, comparing Fast adversarial training (FastAdv) with improved fast adversarial training (FastAdv+) on CIFAR10.

Weakness:
1. FastAdv+ itself does not perform better than PGD adversarial training. When use the FastAdv+ result as starting point then do PGD adversarial training, it outperforms PGD adversarial training. Therefore, FastAdvW (the combined method) is actually more computational expensive than PGD adversarial training.
2. The advantage of Fast adversarial training is that it is less computational expensive than PGD adversarial training so it can scale up to large dataset like ImageNet. However, FastAdvW does not have this advantage anymore.
3. The improvement of FastAdv+ over FastAdv on CIFAR100 and TinyImagenet is marginal.
4. The methods are evaluated under one attack strength ($\epsilon=8/255$ for CIFAR10). It is better to evaluate the methods under different attack strengths.

Clarity:
The paper is clearly written and easy to follow.

Reproducibility:
Details of the algorithm is provided but code is not.

Conclusion:
The method is novel but the contribution is not strong.

---

> ### Author Response · Authors · 2020-11-14
> **Thank you for the valuable comments**
>
> Thank you for the valuable comments. We now respond to individual comments in the following:
>
> 1) We propose FastAdv+ as an improvement upon FastAdv, while FastAdvW as an improvement upon PGD-AT. Please note we only use FastAdv+ in the initial phase (e.g. the first 70 epochs) for FastAdvW and PGD-AT at the later phase (e.g. 70-100 epochs). One can think of this as replacing most of the initial epochs of PGD-AT with FastAdv+, thus always reducing the computational time by a large margin.
>
> 2) Our goal is to replace PGD-AT (not FastAdv) with FastAdvW as it is faster and equivalently good. Although FastAdv and FastAdv+ are faster, they only yield suboptimal performance compared to PGD-AT and FastAdvW. In addition, although we do not have the computational resource to directly experiment on ImageNet, we believe FastAdvW could be used on ImageNet as it is faster than FreeAdv, a method that scales to ImageNet, as shown in Table 2.
>
> 3) We believe the improvement on CIFAR-100 is not marginal, especially considering the improvement in clean accuracy (60.57 vs 55.71). In addition, the advantage of FastAdv+ is not only improved performance but also the ability to run more epochs, which enables many more applications. For example, our proposed FastAdvW is one of them.
>
> 4) We have included the results for various sizes of attacks in the revised paper.
>
> 5) Please find our code in the supplementary materials.

---

### Official Review · AnonReviewer1 · 2020-10-30
**A useful paper needing more theoretical interpretations**

**Rating:** 5
**Confidence:** 4

**Review:**

The authors claimed in this paper that as the most empirically successful approach to defending adversarial examples, PGD-based adversarial training, is computationally inefficient. Fast adversarial training could mitigate this issue by training a model using FGSM attacks initialized with large randomized perturbations, but the underlying reason for its success remains unclear and it may still suffer from catastrophic overfitting. The authors conducted a series of experiments to figure out the key to the success and properties of fast adversarial training. The experimental results showed that fast adversarial training cannot avoid catastrophic overfitting, but could be able to recover from catastrophic overfitting quickly. Based on all of the observations, the authors proposed a simple method to improve fast adversarial training by using PGD attack as training instead of R+FGSM attack (proposed in fast adversarial training) when overfitting happens, or using fast adversarial training as a warmup. The proposed methods could achieve slightly better performance than the current state-of-art approach while reducing the training time significantly.

#####################################################################

Overall, I vote for weak reject (marginally). I like the idea of exploring the properties of adversarial training, the experiments may also be inspiring. But my major concern is that the interpretation about the ‘catastrophic overfitting’ is not clear, and the interpretation about the effectiveness of R-FGSM and PGD against overfitting is also not clear. Hopefully, the authors can address my concern in the rebuttal period.

Pros:
#####################################################################Pros:

1.Attempting to interpret the successful reason for a previous work is interesting. And the exploratory experiments may be inspiring for other researchers.

2. Overall, the paper was well written. All the motivations and conjectures are easy to follow and understand.

3. This paper provides a lot of experiments to show the effectiveness of the proposed methods which appeared slightly better than the SOTA PGD-training while reducing training time significantly.
#####################################################################


Cons:

1. Although the authors attempted to explain the key to the success of fast adversarial training, it might be still not clear theoretically:

(1)	Why R-FGSM and PGD could guide the model to recovery from ‘catastrophic overfitting’, but FGSM could not? Does it mean that stronger attacks could guide the model to recovery?

(2) Why the ‘catastrophic overfitting’ happened a lot of times when using FGSM training, but R-FGSM and PGD could mitigate it? Does it mean that stronger attacks could mitigate it?

2. As I understand, Figure 3(c) should be the result of proposed FastAdv+. From Figure 3(c), it can be observed that there are ‘catastrophic overfitting’ in FastAdv+, but this phenomenon could not be seen from Figure 4. Do you have any idea to explain it?

3. As concerned in my 1.(1) and (2), if weaker attacks lead to more ‘catastrophic overfitting’ and could not guide the model to recovery, why the FastAdvW 4-8 using a weaker attack as a warmup could outperform FastAdv+ and FastAdvW 8-8.

4. Though the proposed methods appear useful, they may be a bit straightforward and have a limited novelty (using PGD attacked samples when ‘catastrophic overfitting’ happens)

---

> ### Author Response · Authors · 2020-11-14
> **Thank you for the valuable comments.**
>
> Thank you for the valuable comments. We now respond to individual comments in the following:
>
> 1-1) The cause of catastrophic overfitting has been studied in Trame`r et al. [1]. Therefore, we only summarized their findings in Section 3. For a more detailed explanation, Figure 1 in their paper has shown the loss surface in the input space when overfitting happens. When a single-step attack is used during adversarial training, the model learns to fool the attacker by making the local gradient a poor approximation to the global loss, which is eps1 in plot b (zoom-in view), while eps2 is a better approximation globally as shown in plot a. The reason R-FGSM helps is that its randomized initialization could jump out of the area where the gradients are misleading. In fact, it is the motivation for introducing R+FGSM in the first place.
> However, since it is randomized, there is a chance it jumps to another area where the gradient is still misleading (essentially Wong et al. [2] show a larger random initialization could mitigate this issue, but we show it still happens). On the other hand,  multi-step updates of PGD attacks allow it to effectively find the correct direction for constructing adversarial examples by making small updates each time and can consistently escape from the misleading area.
>
> 1-2) This also explains why R-FGSM and PGD guide the model recovery from ‘catastrophic overfitting', as they can re-discover the ‘true gradient’ that approximates the global loss. In general, any strong attack that the model cannot fool could do the job, but PGD is the most commonly used one for adversarial training.
>
> 2) In Figures 2 and 3, we plot robust accuracy after every 20 batches, while in figure 3 we plot robust accuracy after each epoch. Since catastrophic overfitting and recovery only happens within a few batches, it is often not observable when we only look at the robust accuracy for each epoch.
>
> 3) Firstly, we use the proposed FastAdv+ in the initial phase, which always guides the model to recovery. The occurrences of catastrophic overfitting do not undermine the final results as long as we make sure the model recovers.
> FastAdvW 4-8 outperforms FastAdv+ because we used PGD adversarial training in the late phase. It outperforms FastAdvW 8-8 because the latter one uses a stronger attack (FGSM with size 8/255) during warmup and causes model overfitting (not catastrophic overfitting but the one discussed in Rice et al. [3]).
>
> 4) We believe being simple and straightforward is the strength of our approach. As long as our approach is effective, its simpleness makes implementation much easier.
>
> [1] Tramèr, Florian, et al. "Ensemble Adversarial Training: Attacks and Defenses." International Conference on Learning Representations. 2018.
>
> [2] Wong, Eric, Leslie Rice, and J. Zico Kolter. "Fast is better than free: Revisiting adversarial training." arXiv preprint arXiv:2001.03994 (2020).
>
> [3] Rice, Leslie, Eric Wong, and J. Zico Kolter. "Overfitting in adversarially robust deep learning." arXiv preprint arXiv:2002.11569 (2020).

---

### Decision · Program_Chairs · 2021-01-07
**Final Decision**

**Decision:**

Reject

**Comment:**

This paper first investigates the behavior (e.g., catastrophic overfitting) of fast adversarial training (FastAdv) through experiments. It finds that the key to its success is the ability to recover from overfitting to weak attacks. Then, it presents a simple fix (FastAdv+) that incorporates PGD adversarial training when catastrophic overfitting is observed. The resulting method is shown to be able to train for a large number of epochs. It also presents a version (FastAdvW) that use the improved fast adversarial training as a warmup of PDG-adversarial training, similar as in previous work. Overall, the analysis is useful and the ideas are valid. The empirical results also show promise. However, the main weakness of such empirical analysis is that it may be sensitive to the settings (e.g., # of epochs, splitting of datasets, …). The authors’ rebuttal also reflected such potential concerns.